# Microorganism-Derived Molecules as Enzyme Inhibitors to Target Alzheimer’s Diseases Pathways

**DOI:** 10.3390/ph16040580

**Published:** 2023-04-12

**Authors:** Thi Hanh Nguyen, San-Lang Wang, Van Bon Nguyen

**Affiliations:** 1Doctoral Program in Applied Sciences, Tamkang University, New Taipei City 25137, Taiwan; nguyenhanh2208.tn@gmail.com; 2Department of Chemistry, Tamkang University, New Taipei City 25137, Taiwan; 3Institute of Biotechnology and Environment, Tay Nguyen University, Buon Ma Thuot 630000, Vietnam

**Keywords:** Alzheimer’s disease, enzyme inhibitors, microbial source, virtual study, ChE, secretase, GSK-3β, MAO, PKC, PDE

## Abstract

Alzheimer’s disease (AD) is the most common form of dementia. It increases the risk of other serious diseases and causes a huge impact on individuals, families, and socioeconomics. AD is a complex multifactorial disease, and current pharmacological therapies are largely based on the inhibition of enzymes involved in the pathogenesis of AD. Natural enzyme inhibitors are the potential sources for targeting AD treatment and are mainly collected from plants, marine organisms, or microorganisms. In particular, microbial sources have many advantages compared to other sources. While several reviews on AD have been reported, most of these previous reviews focused on presenting and discussing the general theory of AD or overviewing enzyme inhibitors from various sources, such as chemical synthesis, plants, and marine organisms, while only a few reviews regarding microbial sources of enzyme inhibitors against AD are available. Currently, multi-targeted drug investigation is a new trend for the potential treatment of AD. However, there is no review that has comprehensively discussed the various kinds of enzyme inhibitors from the microbial source. This review extensively addresses the above-mentioned aspect and simultaneously updates and provides a more comprehensive view of the enzyme targets involved in the pathogenesis of AD. The emerging trend of using in silico studies to discover drugs concerning AD inhibitors from microorganisms and perspectives for further experimental studies are also covered here.

## 1. Introduction

Alzheimer’s disease (AD) is a multifactorial disease featured by the deposition of amyloid-beta (Aβ) plaques and neurofibrillary tangles in the brain, leading to the death of neuronal cells and memory loss [1,2]. AD increases the risk of other serious diseases and has a great impact on individuals, families, and socioeconomics [3]. Currently, there are about 50 million AD patients worldwide, and this number is predicted to double every five years; it is expected to increase to 152 million by 2050 [2,4]. AD is a multifactorial disease with a complex pathophysiology, and to date, the exact etiology has not been elucidated, making the disease difficult to treat [1,2]. To date, there is no cure for AD; instead, there are treatments that improve the symptoms and conditions of the disease [5,6]. The development of compounds can prevent or treat AD by targeting several pathogenic mechanisms [7,8]. According to AD pathogenesis, current pharmacological therapies are mainly based on the inhibition of target enzymes causing AD [9]. However, these traditional drugs only affect a single target, helping to reduce symptoms and disease progression while causing many side effects. Therefore, efforts to find new potential inhibitors are attracting increasing scientific interest.

The sources of natural inhibitors targeting enzyme inhibition for AD treatment are mainly plants, marine organisms, and microorganisms [10,11,12,13]. Of these, the sources of plants and marine organisms all are limited in exploited productivity, while inhibitors from microorganisms have many advantages over others [14]. Natural inhibitors from microorganisms can overcome the disadvantages of synthetic inhibitors in terms of toxicity, while this problem is limited in other natural sources. It is especially easy to control inhibitor production by microbes on a large scale with the advantage of a cost-effective and friendly environment. Thus, numerous inhibitors from microorganisms that target at inhibiting enzymes related to AD treatment were discovered [10,14].

Several review papers related to AD that mainly present general theories of AD have been published [7,15,16,17,18,19,20,21]. Other reviews have also overviewed AD-related inhibitors; however, most of them focused on some common target enzymes, such as acetylcholinesterase [22,23,24,25], β-secretase [26,27], and glycogen synthase kinase-3 [13], and mainly presented the sources of inhibitors through chemical synthesis, plants, or marine organisms [11,12,13,22,23,24,26,28]. Reviews concerning enzyme inhibitors from microbes are still limited [10,14]. The current new trend is the use of multi-targeted drugs for AD treatment; understanding the enzymes involved in the pathogenesis of AD will help in the determination of enzyme targets in related experimental research toward the inhibition of pathogenesis enzymes of AD. Therefore, this review provides a more comprehensive view of different target enzymes in AD management. We present here the sources of inhibitors from microbial sources on various enzymes targeting AD. The emerging trend of using in silico studies for the drug discovery of Alzheimer’s inhibitors from microorganisms and perspectives for further investigation were also presented and discussed in this review work.

## 2. Overview of Enzyme-Associated Pathogenesis Mechanisms of Alzheimer’s Disease

Many target enzymes related to the pathogenesis mechanisms of AD have been recorded (Figure 1). Cholinergic neurotransmission has been shown to be intimately involved in several key psychological processes, such as memory [29]. The cholinergic hypothesis is the earliest hypothesis for AD, which proposes that the cause of the disease is an impaired synthesis of the acetylcholine neurotransmitter induced by acetylcholinesterase (AchE). The enzyme butyrylcholinesterase (BuChE), similar to AchE, also plays an important role in the progression of AD [30,31,32]. One of the main reasons for the resistance of AD to AChE inhibitors is that BuChE acts as a substitute for the loss of AChE in the neurons of patients with AD. Thus, BuChE continues the activity of AChE in cases where AChE is insufficient or inhibited. Therefore, several studies have evaluated the inhibitory activity of both of these enzymes, simultaneously targeting multi-targeted inhibitors [33]. Although other relevant pathophysiological mechanisms have been further investigated in recent years, treatments that improve cholinergic function remain important in the management of patients with AD [34]. Of the five drugs approved by the Food and Drug Administration (FDA) for AD therapy, four are acetylcholinesterase inhibitors (AchEI), and the other is memantine, an N-methyl-D-aspartic acid receptor antagonist. However, these drugs only help reduce the symptoms of dementia while causing many side effects. To date, several new drug applications have been developed to exploit other new targets of the disease. However, effective therapeutic agents to correct the disease have not yet been found. Besides cholinergic reconstitution, researchers are also looking for other AD targets [35]; two major factors in the development of AD are identified as involving insoluble Aβ peptides and abnormal Tau proteins [36].

The amyloid hypothesis suggests the clinical symptoms of the disease are related to the accumulation of insoluble Aβ peptides due to altered transmembrane amyloid precursor protein (APP) processing. Normal APP processing is achieved by cleavage of the precursor APP by α-secretase, followed by γ-secretase to form soluble Aβ. In pathological cases, precursor APP is cleaved by β-secretase and γ-secretase, leading to the formation of insoluble Aβ, contributing to synaptic damage [9]. Thus, many strategies have been tested for the treatment of AD based on this hypothesis by inhibiting β- and γ-secretase, which are responsible for generating Aβ, regulating Aβ incorporation, or enhancing Aβ elimination [37]. Enzyme-related therapies based on this hypothesis involve the inhibition of β- and γ-secretase while also activating α-secretase activity. However, the inhibition of γ-secretase can cause undesirable side effects because γ-secretase has many vital physiological functions. At present, whether the function of γ-secretase in Aβ production can be specifically inhibited without interfering with other important functions of this protease remains unclear [38]. Thus, β-secretase inhibitor candidates are of particular interest to develop, and the majority of AD treatment studies targeting secretase inhibition have focused on this enzyme [39,40,41].

The Tau protein hypothesis is one of the most important hypotheses of AD related to the formation of fibrillary tangles because of the formation of fibrin plexuses by over-phosphorylated Tau proteins, in which glycogen synthase kinase-3 (GSK-3) is the major kinase responsible for phosphorylating Tau proteins [9]. The elevated phosphorylation can be controlled through the inhibition of the enzyme GSK3 [42]. Much effort has been made in the discovery and development of GSK-3 inhibitors, as this is an area of research that has been actively explored by academic centers and pharmaceutical companies. Although numerous GSK-3 inhibitors have been developed, no GSK-3 inhibitor has been placed in the market due to various concerns regarding the non-selective-target activity causing serious side effects. Thus, despite being a potential AD therapeutic target, analytical evaluations using a specific GSK-3 inhibitor still need to be clarified in-depth [43].

In addition to the above hypotheses, other enzyme targets for AD are also of great interest. For instance, monoamine oxidase (MAO), an activated enzyme, plays an important role in the pathogenesis of AD, including the formation of amyloid plaques from Aβ production and accumulation and the formation of neurofibrillary tangles and cognitive impairment due to cholinergic destruction of neurons and disturbances of the cholinergic system [44]. Therefore, MAO inhibitors (MAOIs) can be considered promising therapeutic agents for AD. While the first-generation MAOIs are indistinguishable, second-generation MAOIs preferentially inhibit MAO-A or MAO-B [45]. Clinical trials involving MAO-inactivating drugs have been conducted; nonetheless, MAOIs have many drug interactions that can produce some undesirable side effects [46]. Similar to γ-secretase, the enzyme protein kinase C (PKC) also encountered some inconsistencies in its handling when early studies suggested the pharmacological activation of PKC as a target for the treatment of AD [47,48]. However, the prolonged activation of PKC is also associated with AD pathology; hence, the activation or inhibition of this enzyme for the treatment of AD also needs further consideration and evaluation [49]. Some other target enzymes, such as cyclin-dependent kinase (CDK-5), microtubule affinity regulating kinase (MARK), phosphodiesterase (PDE), and NADH oxidase, are also reported to be related to AD [50]. CDK-5 is a proline-directed serine-threonine protein kinase [51]; it plays a vital role in the physiological development of the central nervous system and phosphorylates several relevant substrates. CDK-5 is activated by its neuron-specific and membrane-localized (p35 and p39) or respective truncated forms (p25 and p29). Enhanced CDK5 activity leads to abnormal hyperphosphorylation or enhanced amyloid production, causing the neurodegenerative pathology of AD [52,53,54]. MARK is a kinase that plays initiating role in Tau abnormality phosphorylation [55]. PDEs are an enzyme family that hydrolyzes the 3′-phosphodiester links in cyclic guanosine monophosphate (cGMP) and cyclic adenosine monophosphate (cAMP) in signal-transduction pathways for the generation of 5′-cyclic nucleotides. PDEs are essential for controlling cell functioning and adjusting the levels of cAMP and cGMP. Abnormal cAMP signaling is related to AD [56]. NADPH oxidase is the main enzyme causing the creation of damaging free radicals that lead to oxidative stress—a general cause in the pathology of neurodegenerative disorders such as AD [57]. Research has shown that the activity of ERK1/2 (the extracellular-signal-regulated protein kinase of the mitogen-activated protein kinase family) is involved in Tau phosphorylation in AD [58]. Therefore, ERK1/2 is tightly implicated in AD pathogenesis, and it has also become a promising therapeutic target. Currently, no effective therapeutic agents are targeting ERK1/2 for the treatment of AD; thus, related studies are still ongoing [59].

Although various kinds of enzymes were discovered involving AD pathology, studies have mainly focused on the inhibition of enzymes commonly hypothesized to participate in the pathogenesis of AD [8]. While other enzymes have not been studied much, until now, five commercial drugs were issued, and of those, two are of natural origin [60]. Four of the drugs are AChE inhibitors (donepezil, rivastigmine, galantamine, and tacrine), and the other is an NMDA receptor antagonist (memantine) [61]. However, these drugs only support treatment to help reduce the symptoms of dementia and also cause many side effects. In 2021, the FDA expedited the approval of an intravenous injection: aducanumab (AduhelmTM) [61]. Drugs in this category may halt clinical degeneration to improve cognitive function and impact individuals with AD [61]. Additionally, studies to find drugs with effective pharmacology in the treatment of AD continue to be carried out. Recently, there were around 868 drugs in different trial stages. However, only 273 drugs are being actively developed by biotech or pharma companies [2].

## 3. Overview of Inhibitors on Target Enzyme for Alzheimer’s Treatment from Microorganisms

### 3.1. Inhibitors on AchE and BuChE from Microorganisms

Some cholinesterase (ChE) inhibitors were discovered in bacteria (the results are presented in Table 1). The red pigment—prodigiosin from *Serratia marcescens* CC17—was recorded as possessing a favorable ability of AchE inhibition with an IC50 value of 640 µg/mL [62]. Some secondary metabolites isolated from actinobacteria were also tested for anti-AChE activity and showed promising activity. A new phenazine collected from *Streptomyces* sp. LB173 showed a favorable effect on AChE, with a low IC50 value of 2.62 µM [63]. Dimeric indole derivatives from *Rubrobacter radiotolerans* marine actinomycetes also presented a favorable impact on AchE, with potential IC50 values [64]. Physostigmine isolated from *Streptomyces* sp. AH-14 was tested for AChE inhibition, and it showed an IC50 value of 41 μM [65]. An oxygen heterocyclic compound—cyclophostin from *Streptomyces lavendulae* NK901093—was assessed for anti-AchE activity and recorded an IC50 value of 7.6 × $10^{-10}$ M [66]. Pyrrole derivatives produced from *Streptomyces* sp. UTMC 1334 were found to be a potential source of anti-AchE with a low IC50 value of 360 µg/mL [67]. A secondary metabolite—nostocarboline from cyanobacteria *Nostoc* 78-12A—was discovered to possess the ability of anti-BuChE, with an IC50 value of 13.2 μM [68]. Overall, research on ChE inhibitors from bacteria is still limited, and focus has mainly been given to the inhibition of AchE.

Besides bacteria, secondary metabolites from fungi were also found to be potential sources of ChE inhibitors. Various ChE inhibitors from fungi are summarized in Table 2. Research by Kim et al. [69] discovered new ChE inhibitors from two species of fungi of the genus *Penicillium* sp., including quinolacacins A1 and A2, of which only quinolacacin A2 exhibits activity on both AchE and BuChE, with respective IC50 values of 19.8 µM and 650 µM. A new meroterpene and a new benzofuran derivative, along with other components from the marine fungal culture *Acremonium persicinum* KUFA 1007 bound to sponges, were tested for AchE and BuChE inhibitory activity; among them, only lumichrome showed inhibition of AchE, with an IC50 of 12.24 M [70]. Some compounds from *Penicillium* sp. FO-4259 were isolated and tested for ChE inhibition [71,72]. Of those, Omura et al. [71] confirmed that a new inhibitor arisugacin was co-isolated with two known substances—territrems B and C—and the new compound exhibited the best activity on both AchE and BuChE, with low IC50 values of 1 nM and 18000 nM, respectively. Otoguro et al. [72] discovered a variety of other metabolites from FO-4259 comprising arisugacins A–H, which showed inhibitory activity against ChE. Among them, four arisugacins, A–D, showed favorable AchE inhibitory activity, with IC50 between 0.001–3.5 nM, and all compounds also showed strong BuChE inhibitory activity, with IC50 values over 30 nM [72]. Xyloketal A from *Xylaria* sp. exhibited AChE inhibition, with an IC50 value of 1.5 × 10^−6^ mol/L [73]. Compound 14-(2′,3′,5′-trihydroxyphenyl) tetradecan-2-ol was obtained from *Chrysosporium* sp. and showed inhibitory potential against three types of AchE enzymes, including AchE from human serum (197 M), rat brains (195 µM), and electric eels (231 µM) [74]. Paecilomide isolated from *Paecilomyces lilacinus* fungus was evaluated for AChE inhibition, and it showed an inhibition rate of 57.5% at 10 mg/mL [75]. Three compounds from marine fungi *Talaromyces* sp. LF458, including talaromycesone A, talaroxanthenone, and AS-186c, presented potent AChE inhibitory activity, with respective IC50 values of 7.49 μM, 1.61 μM, and 2.60 μM [76]. A diterpene from *Syncephalastrum racemosum* fungus showed a favorable effect against AChE, with a low IC50 value above 1 µM [77]. Three compounds were collected from *Aspergillus sydowii* and tested for AChE inhibition, showing IC50 in the range of 1.06–1.24 μmol/mL [78]. Three new compounds belonging to lipopeptide epimers and phthalide glycerol ether from *ochliobolus lunatus* SCSIO41401 (marine-algae-associated fungus) showed favorable anti-AChE activity, with IC50 values in the range of 1.3–2.5 μM [79].

Several compounds from the endophytic fungus *Talaromyces aurantiacus* FL15 were assessed for the inhibition of ChE, and all compounds showed weaker activity on BuChE than AChE, with IC50 of over 100 µM [80]. Besides that, three asterric acid derivatives presented a favorable effect on AChE, with low IC50 values from 20.1–66.7 µM [80]. Avertoxin B isolated from the endophytic fungus *Aspergillus versicolor* Y10 was tested for human-AChE inhibitory activity, and its IC50 value reached 14.9 μM [81]. The anti-ChE activity of benzopyranones compounds from *Hyalodendriella* sp. Ponipodef12 endophytic fungus was studied, and the IC50 values were in the range of 103.7–135.52 µg/mL [82]. Some secondary metabolites from various mangrove endophytic fungi were isolated and tested for AChE inhibitory activity [83,84]. Compounds from *Penicillium* sp. sk5GW1L, including arigsugacin I, arigsugacins F, and territrem B, showed favorable activity, with respective IC50 values of 0.64 µM, 0.37 µM, and 7.03 µM [83]. Two terphenyls, including 3″-deoxy-6′-O-desmethylcandidusin B and 6′-O-desmethylcandidusin B from the fungus *Penicillium chermesinum* (ZH4-E2), inhibited AChE with IC50 values of 7.8 μM and 5.2 μM, respectively [84]. A new sesquiterpenoid—colletotrichine B from the fungal endophyte *Colletotrichum gloeosporioides*, GT-7—showed AChE inhibition activity, with an IC50 value of 38 μg/mL [85]. Koninginin T—a new polyketide isolated from the endophytic fungus *Phomopsis stipata*—inhibited AChE at a concentration of 10.0 μg [86].

Huperzine A is a natural and potential AChE inhibitor from plants; this compound is produced via fermentation for large-scale production [87]. Its relatively favorable anti-AChE activity was also reported (IC50 value of 0.6 μM) [87]. Two new picoline-derived meroterpenoids from *Amphichorda feline*, including amphichoterpenoids D and E, were assessed for AChE inhibition, and their IC50 reached 12.5 µM and 11.6 µM, respectively [88]. Two cyclohexanoids from *Saccharicola* sp., an endophytic fungus, also recorded AChE inhibition activity, with IC50 values of 0.026 and 0.053 mg/mL [89]. Various metabolites from the *Aspergillus terreus* fungus were isolated and tested for ChE inhibition activity [90,91,92,93]. Bioactive compounds from *A*. *terreus* (No. GX7-3B) also recorded remarkable AChE inhibition, with IC50 values in the range of 1.89–6.71 µM [90]. Territrem B, collected from *A*. *terreus*, inhibited AChE with an IC50 value of 7.6 μM [91]. A recent study in 2022 by Cui et al. [92] tested the ability of butenolide derivatives from the fungus *A*. *terreus* to inhibit AChE and BuChE. Most of these compounds showed better activity on BuChE than AChE, with the highest inhibition rate of 76%, while these values for AChE were only under 23.2%. Some compounds were indicated to be competitive inhibitors of BuChE, with K_i_ values in the range of 12.3–38.2 µM [92]. Four terreusterpenes, A–D, were tested for AChE inhibition; among them, the greatest activity reached 8.86 μM for terreusterpenes D, while IC50 values of compounds A–C were over 40 µM [93]. The AChE and BuChE inhibition by 5-hydroxy-2-methyl-chroman-4-one from endogenous lichen fungi *Daldinia fissa* was tested and showed IC50 values over 40 µM and were found to be a reversible competitive inhibitor of MAO-B, with a K_i_ value of 0.896 µM [94]. Some territrem and butyrolactone derivatives from the fungus *Aspergillus terreus* showed a high ability of anti-AchE, with I50 values from 4.2–5700 nM [95]. 

In total, 87 ChE inhibitors from microorganisms were presented in this review (Figure 2 and Figure 3). Of these, there were fifteen territrem derivatives (**13**–**23**, **84**–**87**), fifteen butenolide derivatives (**64**–**78**), thirteen terpenoids (**10**, **30**, **49**, **50**, **53**, **56**, **57**, **59**, **79**–**82**), seven alkaloids (**1**, **5**, **7**–**9**, **26**, **55**), five ether derivatives (**29**, **36**–**39**), four chromene derivatives (**45**–**48**), three diketopiperazines (**31**–**33**), three polyketides (**54**,**61**,**83**), three hydroxyanthraquinones (**40**–**42**), two dimeric indole derivatives (**3**, **4**), two terphenyls (**51**, **52**), two lipopeptide epimers (**34**, **35**), one phenazine (**2**), one bicyclic enolphosphates (**6**), one benzopyran derivatives (**11**), one flavin (**12**), one ketal (**24**), one fatty alcohol (**25**), one oxaphenalenone dimer (**27**), one isopentenyl xanthenone (**28**), one benzophenone (**43**), one prenyl asteltoxin derivatives (**44**), one cyclohexanoid (**58**), one naphthoquinone (**60**), one steroid (**62**), and one trimeric cyclodepsipeptide (**63**). Based on the anti-ChE activity results, almost all territrem derivatives showed a more efficient inhibitory effect than other families, with IC50 values in the range of 0.001–26,000 nM [71,72,95]. Additionally, there are also differences in bioactivity among these territrem derivatives, which Otoguro et al. [71] indicated may be due to the structure–activity relationship of territrem derivatives. The structures of arisugacin A **(16)**, arisugacin B **(17)**, territrem B (**14**), and territrem C (**15**) are only different in the substituents on their aromatic moiety, while the remaining arisugacins C–H (**18**–**23**) differ in structure from the two in rings A and B. The activity of (**18**) and (**19**) are lower than (**17**) around 97 and 136 times; as such, it also showed the key role of the enone moiety for anti-ChE activity. Compounds (**20**) and (**22**) presented very low anti-AChE activity, which may be related to the role of the 4a-OH and ketone moiety of rings A for AChE inhibition [71]. Nong et al. [95] also reported that the enone group at the A-ring plays a role in the AChE inhibition capacity of territrems and is also related to linking with the active site of AChE. Some other reports also indicated the relationship between structure and bioactivity. Ohlendorf et al. [63] found that 1,6-hydroxylation in the phenazine core structure is important for AChE inhibition. Meng et al. [82] suggested that chlorine substitution at position 2 may contribute to the anti-AChE activity of benzopyranones compounds. Understanding the structure–bioactivity relationship for modifying the structure may lead to the improvement of the anti-ChE activity of inhibitor compounds [77]. The biotransformation of the trachyloban-19-oic acid (**30**) skeleton improved the capacity of AChE inhibition [77]. The result of this report [77] showed that C-17 oxidation in the trachylobane diterpene skeleton significantly enhanced anti-AChE activity. Additionally, the combination of the rearrangement of trachylobane to a kaurane skeleton with C-17 oxidation also improved the bioactivity. This is a great research direction for further studies to enhance bioactivity via structural optimization. However, there have only been a few reported studies concerning modifying inhibitor compounds originating from microbes for enhancing the AChE inhibitory effect.

Overall, ChE inhibitors were isolated from various microorganism sources such as bacteria, actinobacteria, cyanobacteria, and fungi. Among them, inhibitors from fungi have attracted more attention and have many related reports. Additionally, most studies have mainly focused on assessing AChE inhibition, while only a little research has been conducted on both enzymes, although BuChE was also mentioned as a vital enzyme related to pathogenic mechanisms of AD. Thus, besides ChE inhibitors from fungi, more inhibitors from other sources such as bacteria still need to be identified, as this is also a potential source with advantages in the large-scale and effective cost of production. Moreover, the evaluation of the inhibition of both AChE and BuChE is also very effective in controlling the effects of AD due to the shortage of neurotransmitters. Thus, finding inhibitors with the capacity to inhibit both enzymes will be more effective in supporting AD treatment.

### 3.2. Inhibitors of Other Targeted Enzymes of AD from Microorganisms

Besides ChE inhibition, the control of some other target enzymes of AD also plays an important role in AD management. Inhibitors of other target enzymes of AD from microorganisms are summarized in Table 3. Based on the amyloid hypothesis, inhibitors of β-secretase (BACE1)—an amyloid precursor proteolytic enzyme at position 1 beta (BACE)—are also potential candidates for the treatment of AD. Natural sources of BACE1 inhibitors are mainly obtained from plants [40,96], large fungi, marine organisms, and algae [41]. A few studies have investigated secondary compounds from some fungal strains [93,97,98,99,100]. The inhibition activity of extracts from fungal endophytic *Cytospora rhizophorae* against BACE1 was evaluated. Among them, four extracts were the most effective, with IC50 values under 3.0 μg/mL [97]. Secondary metabolites from the fungus *A*. *terreus* have been isolated, and their BACE1 inhibitory activity was evaluated [93,98]. Four terreusterpenes, A–D, were tested for BACE1 inhibitory activity, and among them, the activity of compound C reached over 40 µM, while the remaining compounds A, B, and D showed a favorable effect, with IC50 values of 5.98, 11.42, and 1.91 µM, respectively [93]. Among some terpenoids from *A. terreus*, three new asperterpene compounds, including E, F, and J, showed significant BACE1 inhibition, with respective IC50 values of 13.3, 5.9, and 31.7 μM [98]. Several sesquiterpenoid metabolites from *Phomopsis* sp. TJ507A endogenous fungal species were investigated, showing BACE1 inhibitory activity in the range of 19–44% at a 40 μM concentration [99]. Hispidin from the *Phellinus linteus* fungus also showed a favorable inhibitory effect, with a low IC50 value of 4.9 μM [100]. Daedalols C, isolated from fungal *Daedalea* sp., showed favorable BACE1 inhibition, with an IC50 value of 14.2 μM [101].

Natural sources of MAOIs have also been studied, including plants, animals, marine organisms, and microorganisms [102,103,104,105]. Natural inhibitory compounds from microorganisms are mainly obtained from fungi and bacteria (Table 3). Two piloquinone derivatives isolated from *Streptomyces* sp. CNQ-027 was tested for MAO-A and MAO-B inhibitory activity, and both inhibitors exhibited better inhibition on MAO-B. Among both, compound 4,7-dihydroxy-3-methyl-2-(4-methyl-1-oxopentyl)-6H-dibenzo[b,d] pyran-6-one showed better activity, with IC50 values of 6.47 μM (MAO-A) and 1.21 μM (MAO-B), and it was found to act as a competitive inhibitor of MAO-A and -B, with respective K_i_ values of 0.573 and 0.248 µM [103]. Compound (Z)-N-(4-hydroxystyryl) formamide from the endophytic fungus *Talaromyces* sp. LGT-2 exhibited favorable anti-MAO activity, with an IC50 reaching 61 μM [106]. Two species of fungi, *Emericella navahoensis* [107] and *Talaromyced luteus* [108], were tested for their ability to inhibit MAO and also MAO-A and MAO-B of secondary metabolites. From *E. navahoensis*, three compounds were obtained, including norsolorinic acid, averufin, and 6, 7, 8-trihydroxy-3-methylisocoumarin, showing an anti-MAO effect, with respective IC50 values of 0.3 μM, 54.4 μM, and 817.3 μM. Further assessment of the efficacy of norsolorinic acid showed that it inhibited MAO-A and MAO-B, with IC50 values of 0.4 and 0.32 μM in rat livers and IC50 values of 4 μM and 0.59 μM in rat brains, respectively [107]. Compound TL-1 from *T. luteus* inhibited MAO with an IC50 of 6.6 μM and MAO-A and -B in rat livers with IC50s of 43 μM and 12 μM, respectively, and respective IC50s of 600 μM and 4 μM in rat brains [108]. The inhibition of MAO activity of three anithiactins from *Streptomyces* sp. was assessed, and anithiactin A showed the highest activity, with IC50 values of 13 μM (MAO-A) and 183 μM (MAO-B). Moreover, anithiactin A was also found to be a reversible competitive inhibitor of MAO-A, with a K_i_ value of 1.84 µM [109]. A recent study by Jeong et al. [110] evaluated the inhibition of the human-MAO activity of (S)-5-methylmellein from the endogenous fungus *Rosellinia corticium* and found that it could inhibit MAO-A and MAO-B with low IC50 values of 5.31 µM and 9.15 µM, respectively. Furthermore, methylmellein acted as a reversible competitive inhibitor of hMAO-A, with a K_i_ value of 2.45 μM [110]. The MAO inhibition of 5-hydroxy-2-methyl-chroman-4-one from endogenous lichen fungi *Daldinia fissa* showed IC50 values of 13.9 μM (MAO-A) and 3.2 μM (MAO-B) [94].

Based on the Tau hypothesis, the inhibition of GSK3 is also a potential target for AD therapy [42]. Marine organisms, especially marine invertebrates, are important sources of novel GSK3 inhibitors [42], although the literature on exploiting the potential of GSK3 inhibitors from microorganisms is very limited. Research by Wiese et al. [111] isolated three new GSK-3β inhibitors from a marine fungus *Aspergillus* sp., in which alternariol inhibited most effectively, with an IC50 value of 0.13 μM, followed by alternariol-9-methyl ether with IC50 of 0.20 μM and finally pannorin with an IC50 of 0.35 μM. Biscogniauxone, a new isopyrrolonaphthoquinone isolated from the marine deep-sea fungus *Biscogniauxia mediterranea*, showed high anti-GSK-3β activity with an IC50 value of 8.04 μM [112]. Type 4 phosphodiesterase (PDE4), including four subtypes (4A, 4B, 4C, and 4D), catalyzes the hydrolysis of cAMP, and these enzymes were found to be associated with neurological diseases [113]. Three compounds were obtained from an ascomycete fungus *Phoma* sp. used for the evaluation of PDE4B inhibition [114]. The activity of two benzoquinones, including betulinan A and betulinan C, showed favorable effects, with respective IC50 values of 44 μM and 17 μM, and the IC50 value of the remaining terphenyl compound reached 31 μM [114]. Some metabolites of the coral-associated fungus *Aspergillus* sp. ITBBc1 were assessed for anti-PDE4D activity [115]. At a concentration of 5 µM, sanshamycin C showed the highest activity, with an inhibition rate of 49.4%, while the other compounds had weaker inhibitory effects (4.8–23.2%) [115]. Some metabolites from *Streptomyces* were isolated and used for the evaluation of PKC inhibition. Four indolocarbazoles alkaloids, including 30-epi-k252d (144), 20,40-*epi*-k252d (145), K252d (146), and sreptocarbazoles C (143) from *Streptomyces* sp. A65, demonstrated inhibition against PKC, with IC50 values of 0.25, 0.35, 0.97, and >20 µM, respectively [116]. Some metabolites (compounds 147, 148, 149, and 151) from *Streptomyces* sp. A68 effectively inhibited the activity of PKCα, with low IC50 values in the range of 0.17–1.32 µM, while compound 150 showed weak activity, with an IC50 value of more than 20 µM [117]. Other indolocarbazoles from *Streptomyces* sp. DT-A61 also presented favorable effects on PKCα, with IC50 values under 3.2 µM [118]. Compound 12-N-methyl-k252c from the A22 strain showed favorable anti-PKC activity, with an IC50 value of 1.84 μM [119].

**Table 3 pharmaceuticals-16-00580-t003:** Inhibitors of other target enzymes in Alzheimer’s disease from microorganisms.

Strain	Compound	Bioactivity (IC50)	Ref.
Anti-BACE1			
*Cytospora rhizophorae*	Extracts	<3.0 μg/mL	[97]
*Aspergillus terreus*	Terreusterpene A (**79**)	5.98 µM	[93]
	Terreusterpene B (**80**)	11.42 µM	
	Terreusterpene C (**81**)	40 µM	
	Terreusterpene D (**82**)	1.91 µM	
*Aspergillus terreus*	Asperterpene E (**88**)	13.3 μM	[98]
	Asperterpene F (**89**)	5.9 μM	
	Asperterpene J (**90**)	31.7 μM	
	Asperterpene D (**91**)	>50 μM	
	Asperterpene G (**92**)	>50 μM	
	Asperterpene H (**93**)	>50 μM	
	Asperterpene I (**94**)	>50 μM	
	Asperterpene K (**95**)	>50 μM	
	Asperterpene L (**96**)	>50 μM	
	Asperterpene M (**97**)	>50 μM	
	Terretonin D (**98**)	>50 μM	
	Terretonin G (**99**)	>50 μM	
*Phomopsis* sp. TJ507A	Phomophyllin A (**100**)	IR = 44%	[99]
	Phomophyllin B (**101**)	IR = 35%	
	Phomophyllin C (**102**)	IR = 19%	
	Phomophyllin D (**103**)	IR = 40%	
	Phomophyllin E (**104**)	IR = 37%	
	Phomophyllin F (**105**)	IR = 38%	
	Phomophyllin G (**106**)	IR = 25%	
	Phomophyllin I (**107**)	IR = 39%	
	Phomophyllin L (**108**)	IR = 4%	
	Phomophyllin M (**109**)	IR = 5%	
	Granulone B (**110**)	IR = 6%	
	Radulone B (**111**)	IR = 39%	
	2-(2,2,4,6-tetramethylindan-5-yl)ethanol (**112**)	IR < 1%	
	Pterosin Z (**113**)	IR < 1%	
	Onitin (**114**)	IR = 42%	
	7-hydroxy-10-oxodehydrodihydrobotrydial (**115**)	IR = 40%	
*Phellinus linteus*	Hispidin (**116**)	4.9 μM	[100]
*Daedalea* sp.	Daedalol C (**117**)	14.2 μM	[101]
Anti-MAO			
*Daldinia fissa*	5-hydroxy-2-methyl-chroman-4-on (**83**)	13.9 μM (MAO-A) 3.2 μM/Ki:0.896 µM (MAO-B)	[94]
*Streptomyces* sp. CNQ-027	4,7-dihydroxy-3-methyl-2-(4-methyl-1-oxopentyl)-6H-dibenzo[b,d]pyran-6-one (**118**)	6.47 μM (MAO-A) K_i_ = 0.573 µM 1.21 μM (MAO-B) K_i_ = 0.248 µM	[103]
	1,8-dihydroxy-2-methyl-3-(4-methyl-1-oxopentyl)-9,10-phenanthrenedione (**119**)	> 80 μM (MAO-A) 14.5 μM (MAO-B)	
*Talaromyces* sp. LGT-2	(Z)-*N*-(4-hydroxystyryl)formamide (**120**)	61 μM	[106]
*Emericella navahoensis*	Norsolorinic acid (**121**)	0.3 μM	[107]
	Averufin (**122**)	54.4 μM	
	6, 7, 8-trihydroxy-3-methylisocoumarin (**123**)	817.3 μM	
*Talaromyced luteus*	TL-1 (**124**)	6.6 μM	[108]
*Streptomyces* sp.	Anithiactin A (**125**)	13 μM (MAO-A) K_i_ = 1.84 µM 183 μM (MAO-B)	[109]
	Anithiactin B (**126**)	>85 μM (MAO-A) - (MAO-B)	
	Anithiactin C (**127**)	>170 μM (MAO-A) >170 μM (MAO-B)	
*Rosellinia corticium*	(S)-5-methylmellein (**128**)	5.31 μM (MAO-A) K_i_ = 2.45 μM 9.15 μM (MAO-B)	[110]
Anti-GSK3			
*Aspergillus* sp.	Pannorin (**129**)	0.35 μM	[111]
	Alternariol (**130**)	0.13μM	
	Alternariol-9-methylether (**131**)	0.20 μM	
*Biscogniauxia mediterranea*	Biscogniauxone (**132**)	8.04 μM	[112]
Anti- PDE			
*Phoma* sp.	Betulinan A (**133**)	44 μM	[114]
	BTH-II0204-207:A (**134**)	31 μM	
	Betulinan C (**135**)	17 μM	
*Aspergillus* sp. ITBBc1	Sanshamycin A (**136**)	IR = 4.8%	[115]
	Sanshamycin B (**137**)	IR = 6.7%	
	Sanshamycin C (**138**)	IR = 49.4%	
	Sanshamycin D (**139**)	IR = 12.4%	
	Sanshamycin E (**140**)	IR = 5.1%	
	Terphenyllin (**141**)	IR = 12.8%	
	3-hydroxyterphenyllin (**142**)	IR = 23.2%	
	Anti-PKC		
*Streptomyces* sp. A65	Streptocarbazoles C (**143**)	>20 µM	[116]
	30-*epi*-k252d (**144**)	0.25 µM	
	20,40-*epi*-k252d (**145**)	0.35 µM	
	K252d (**146**)	0.97 µM	
*Streptomyces* sp. A68	3′-epi-*N*-Acetyl-holyrine A (**147**)	0.17 µM	[117]
	3′-*N*-Acetyl-holyrine A (**148**)	0.91 µM	
	3′-*N*-Formyl-holyrine A (**149**)	1.04 µM	
	Eudesm-4(15), 7-diene9α- hydroxy-11-amino-benzoicacid (**150**)	>20 µM	
	(9R, 22R)-bisphenol A bis (9, 22- hydroxy-10, 23-anthranilicacid-propyl) ether (**151**)	1.32 µM	
*Streptomyces* sp. DT-A61	9-hydroxyk252c (**152**)	0.98 µM	[118]
	3-hydroxy-k252c (**153**)	3.2 µM	
	3-hydroxy-7- methoxy-k252c (**154**)	1.4 µM	
	9-hydroxy-3′-*N*-acetylholyrine A (**155**)	0.097 µM	
	3-hydroxy-3′-*N*-acetylholyrine A (**156**)	0.46 µM	
	3- hydroxyholyrine A (**157**)	0.079 µM	
	3′-*O*-demethyl-4′-*N*-demethyl-4′-*N*-acetyl-4′-epi-staurosporine (**158**)	0.092 µM	
	Streptocarbazole D (**159**)	2.1 µM	
	Streptocarbazole E (**160**)	1.4 µM	
*Streptomyces* sp. A22	12-*N*-methyl-k252c (**161**)	1.84 μM	[119]

IR—Inhibition rate (-) Inactive.

The chemical structures of inhibitors for other enzyme targets of AD were described in Figure 4. Among twenty-five BACE1 inhibitors (**79**–**82**, **88**–**106**, **108**), almost all these inhibitors belong to terpenoids; only Phomophyllin I (**107**) is a polyphenol compound (**107**). Some structure–bioactivity relationships were indicated. In a report by Qi et al. [93], a new terpenoid with a 4-hydroxy-3-methyl gamma lactone fragment named terreusterpene D (**82**) showed higher anti-BACE1 activity than the three remaining terpenoids. In the work by Qi et al. [98], three meroterpenoids showed better anti-BACE1 capacity compared to others due to possessing cis-fused A/B rings. Research by Sorribas et al. [101] found that among three triterpenes from the *Daedalea* sp. fungus, only one compound—Daedalols C—showed anti-BACE1 activity. It possesses the terminal electrophilic epoxide and may form a covalent linkage with BACE1, while the compound lacks this position without bioactivity. The result of a report by Yamazaki et al. [107] indicated that the presence of an anthraquinone skeleton and a side chain of about 6C with a conjugated π electron system at the α-position might play a role in MAO inhibition. In another work, anithiactin A showed higher anti-MAO-A activity than two others, possibly due to the hydrophobic methyl substituent in this compound playing a vital role in the MAO-A inhibition [109]. Three isocoumarin compounds from the *Aspergillus* fungus commonly contain a highly oxygenated benzocoumarin core structure that is suggested to have a role in efficient GSK-3β inhibition [111]. Research by Guo et al. [115] recorded that sanshamycin C showed favorable PDE4D inhibition compared to others due to the two fused six-membered rings and the hydroxyl group at C-2′ in this compound. In a study conducted by Qin et al. [117], the abnormal absolute configuration of a 3′-epi-N-Acetyl-holyrine A compound at C-3′ was possibly related to PKC inhibitory activity. Eight of nine compounds from *Streptomyces* sp. DT-A61 possessed a hydroxy group at either the C-3 or C-9 positions, which differs from other reported natural indolocarbazoles [118]. This set of compounds has provided a small structure−bioactivity relationship, while the compounds with the sugar moiety to the K252c unit showed more potency than those without sugar moiety or compounds possessing only a single attachment of the sugar to the aromatic aglycone. The aspect relationship between the chemical structure and bioactivity of natural compounds remains largely unexplored. This can be very useful in synthesizing potential compounds based on the chemical framework of natural compounds with the aim of enhancing medical effects and safety. This issue is recommended for future research.

In general, most of the research exploring microbial inhibitors for AD treatment targeting has mainly focused on ChEs, and the exploitation of other enzymes is still limited. Compared with natural sources of inhibitors from plants, the research on microbial AD targeting inhibitors is still very limited, with most of the studies focusing on exploiting the secondary compounds mainly from fungi, while very few natural inhibitors from bacteria have been examined. Until now, apart from the five commercial drugs, no new drugs have been developed and introduced to treat AD. The rate of clinical trials has been designed based on each different hypothesis of AD. Of these, the three common hypotheses with the most trials are the amyloid hypothesis (22.3%), the cholinergic hypothesis (19.0%), and the Tau hypothesis (12.7%) [8]. Clinical studies have shown that many single-targeted therapies are not successful in treating the symptoms or progression of multifactorial AD [120] and that the combination of single drugs is more likely to cause drug resistance and side effects [121]. In particular, over 95% of all AD cases are sporadic; evidence also indicates that sporadic AD is complex and involves multiple disease mechanisms [122]. Therefore, the current trend is to develop drugs with multi-target effects of inhibiting multiple enzymes or modulating biosynthetic pathways implicated in dementia and AD [123].

## 4. Trends in Using Virtual Screening to Discover Potential Alzheimer’s Inhibitors from Microorganisms

Experimental screening for potential drugs is a long and costly process [124]. Recently, virtual screening has become an efficient and accessible tool for drug discovery and involves searching a library of small molecules to identify those structures most likely to bind to a drug target, typically protein receptors or enzymes. Virtual screening plays an important role in helping in the design, optimization, and development of novel drugs, reducing in vivo trials in drug discovery and redefining the effects of known drugs. This technology is also widely applied in the search for potential inhibitors related to AD treatment targets [62,125,126,127,128]. According to this trend, some studies have used docking studies to evaluate potential inhibitors from microorganisms for the treatment of AD, and the results are summarized in Table 4. In docking simulations, the binding energy is considered the main indicator to compare the inhibitory activities of inhibitors toward the target enzyme. The lower the docking score (DS) of an inhibitor, the greater its inhibitory capacity. When the docking score (DS) of a ligand interacts with and binds to an enzyme with binding energy lower than −3.20 kcal/mol, it suggests that the ligand possesses a favorable enzyme binding ability [129,130]. Furthermore, the root mean squared deviation (RMSD) is also considered an important indicator in the docking study. An RMSD under 2.0 Å is widely accepted; when this value exceeds 3.0 Å, the predicted inhibition is negligible [130,131].

The red pigment (prodigiosin-PG) from *S. marcescens* CC17 was evaluated for anti-AChE via docking study at two states of cation-PG and neutral-PG [62]. These ligands (inhibitors) showed effective interaction with AChE at the active zone containing 31 amino acids, with low DS values of −12.3 kcal/mol (Cation-PG) and −11.1 kcal/mol (Neutral-PG), and RMSD values under 2.0 Å. They also bonded with the enzyme, with up to 3–6 bonds, while the control compound only showed one bond. Cation-PG bonded to AChE via the interaction of some amino acids, including Asp326, Asp393, and Lys325, and six bonds were created. In contrast, neutral-PG showed lower binding energy and was linked with Trp84 and Gly118, creating three bonds [62]. Two inhibitor compounds from *B. veleznesis* RB.EK7 also presented potential linking with AChE, with low DS values (−7.0 and −6.89 kcal/mol) and RMSD values under 1.36 Å, and they could form 3–4 linkages with AChE [132]. Among them, thymine formed four linkages, including two H-acceptors, one pi-H, and one H-donor at the active sites with some prominent amino acids (Asp182, Lys5, Asn183, and Trp179). The remaining compounds interacted with AChE at the active site via three linkages (one H-donor and two H-acceptors) with some amino acids such as Met175, Phe35, and Lys51 [132]. The binding interaction of compounds from the fungus *A. felina* with AChE was investigated via docking analysis [91]. Residues with interaction energies under −1 kcal/mol are found to be essential for the recognition and complexing of ligands. The interaction of amphichoterpenoid D and E with AChE is similar; with a DS value of −9.3 kcal/mol, the interaction involves three hydrogen bonds and two interacting residues. The DS value of (+)amphichoterpenoids was −7.9 kcal/mol, with two hydrogen bonds and two interaction residues, while the DS remaining compound without hydrogen bonds was –6.8 kcal/mol [88]. Butenolide compounds from the *A. sterreus* fungus showed different interactions with BuChE [92]. The total binding free energy of butyrolactone I and butyrolactone VII was −41.28 kcal/mol and −48.85 kcal/mol. Both compounds were linked with BuChE by the same interactions, such as a π–amide stacked with Gly116, a π–π T-shaped linkage with Trp82, a π-σ interaction with Trp231, and π–alkyl or alkyl interactions with Val288, Leu286, and Phe398. Additionally, each compound also interacted with BuChE through individual bonds [92]. The interaction of (S)-5-methylmellein (SMM) from the *R. corticium* fungus and its isomer (R)-5-methylmellein (RMM) with MAO-A and MAO-B of humans was demonstrated [110]. The binding score of SMM reached −6.8 kcal/mol for MAO-A and -6.4 kcal/mol for MAO-B, while these values for its isomer were −6.6 kcal/mol and −5.2 kcal/mol, respectively. Additionally, both compounds interacted with H-MAO-A, forming hydrogen bonds with Phe208 residue for SMM and with Asn181 residue for RMM [110]. The in silico pharmacokinetic result showed that SMM did not violate Lipinski’s rule of five—an important rule in drug development—and presented a high blood–brain barrier permeability and gastrointestinal absorption. 5-hydroxy-2-methyl-chroman-4-one (HMC) isolated from *Daldinia fissa* fungus was evaluated for the interaction with human-MAO via docking stimulation [94]; HMC possessed a better bonding affinity with MAO-B (−7.3 kcal/mol) than MAO-A (−6.1 kcal/mol). It was linked with Cys172 of MAO- B through hydrogen bonding, while it had no interaction with MAO-A. Additionally, the analysis of the pharmacokinetics of this compound using SwissADME’s web tool indicated that it possesses high gastrointestinal absorption and can possibly cross the blood–brain barrier; moreover, it does not inhibit cytochrome P450. It was also predicted to have no violations of Lipinski’s rule of five in the Lipinski parameters analysis [94]. The interaction of three compounds from *Phoma* sp. with PDE4B was predicted through a docking study [114]. Among them, betulinan C showed the highest DS value of −8.732 kcal/mol, and the DS values of the other two compounds, including betulinan A and BTH-II0204-207: A, were −8.071 kcal/mol and −8.277 kcal/mol, respectively. Favorable π interactions with Phe446 were found in all compounds, while the hydrogen bond with Gln443 was only indicated in the most active compound [114].

Among these compounds for respective enzyme targets, as summarized in Table 4, the effects of two compounds ((R)-5-methylmellein, amphichoterpenoid A) were not reported via in vitro assays. Four compounds, including butyrolactone I, butyrolactone VII, thymine, and hexahydropyrrolo [1,2-a]pyra-zine-1,4-dione, were accessed via in vitro assay and showed moderate enzyme inhibition, with an inhibition rate of approximately 70%. Other compounds demonstrated favorable effects via in vitro tests. Of these, prodigiosin, amphichoterpenoid D, and amphichoterpenoid E inhibited AChE, with low IC50 values of 640 µg/mL, 12.5 µM, and 11.6 µM, respectively. Both 5-hydroxy-2-methyl-chroman-4-one and (S)-5-methylmellein showed inhibitory efficacy against both MAO-A and MAO-B, with IC50 values in the range of 3.2–13.9 μM. Three compounds, including betulinan A, BTH-II0204-207:A, and betulinan C isolated from *Phoma* sp., were tested for inhibition against PDE4B and also showed favorable effects, with IC50 values of 44, 31, and 17 μM, respectively.

Efforts to develop drugs to treat AD are still ongoing [11,61,123,133]. Natural inhibitors targeting AD treatment are considered promising, efficient, and safe sources. However, in fact, screening to discover potential inhibitors by experimenting is not easily conducted on a large number of samples, as it requires a lot of effort and cost for purification and testing bioactivity. Instead, virtual screening has many advantages in screening and predicting potential inhibitors based on the simulation, and finally, the samples with the most potential can be reconfirmed easily by experiments. This tool can also optimize the chemical structure based on the structure of potential natural compounds to customize the structure according to the desired properties, thereby supporting the design of potential drugs with optimal efficacy [134]. However, only a few studies related to virtual simulation for screening or predicting the interaction of inhibitors from microorganisms with the potential of targeting enzymes involved in the pathogenesis of AD have been conducted. Thus, more research exploration on this topic will be needed in the future.

## 5. Conclusions and Perspectives

AD is known to be a common cause of dementia. Some enzymes are considered related to the pathogenesis of AD. In this article, some related enzymes, such as AchE, BuChE, secretase, GSK-3 beta, MAO, PKC, CDK, microtubule affinity regulating kinase, phosphodiesterase, NADH oxidase, and ERK1/2, were mentioned. Additionally, the need to find new inhibitors from safe, natural sources for these enzymes is increasing. In total, 161 inhibitors isolated from microbial sources with inhibitory effects on some of the above enzymes have been synthesized. These results show that many potential inhibitory compounds from microbial sources have been discovered, and a wide range of inhibitors from fungi has been recorded, although there has been little exploitation from other microbial sources. The application of virtual simulations in the evaluation and search of potential inhibitors and identification of the interaction of inhibitors with target enzymes has also often been published. The exploitation of inhibitors from microbial sources to target AD is a potential direction, especially capturing the trend of developing multi-targeted inhibitors in the treatment of multifactorial diseases such as AD.

In the future, many related research directions still need to be further explored. Besides researching the exploitation of inhibitors from fungi, it is advisable to intensify the exploratory studies on other microorganisms, such as bacteria or actinomycetes, to enrich the potential inhibitors for AD from natural sources. Most of the new studies have only focused on investigating inhibitors for some common AD-related enzymes such as ChE; therefore, it is necessary to expand the investigation to the inhibition of other enzymes. The recent trend in the development of AD drugs is multi-targeting, so it is recommended to combine the evaluation of the inhibition of many different enzymes involved in AD towards exploiting molecules capable of inhibiting multi-target enzymes. Furthermore, the integration of virtual screening to identify potential inhibitors from this natural source is also a potential research direction to support future drug development for AD.

## Figures and Tables

**Figure 1 pharmaceuticals-16-00580-f001:**
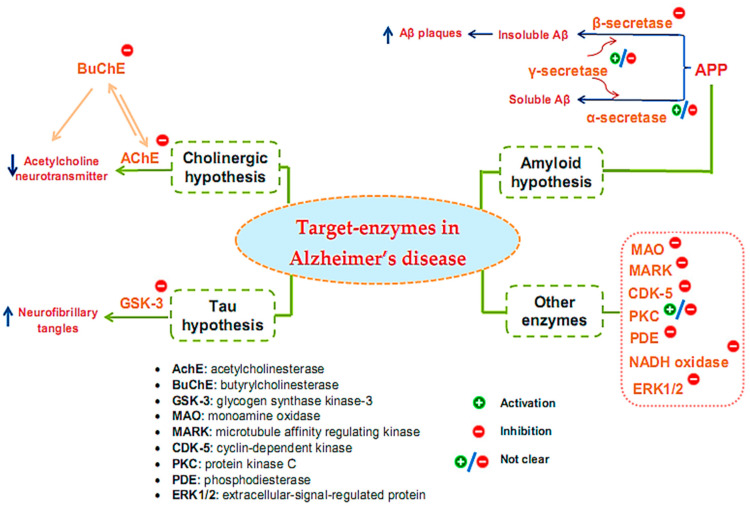
Target enzymes related to the pathogenesis of Alzheimer’s disease.

**Figure 2 pharmaceuticals-16-00580-f002:**
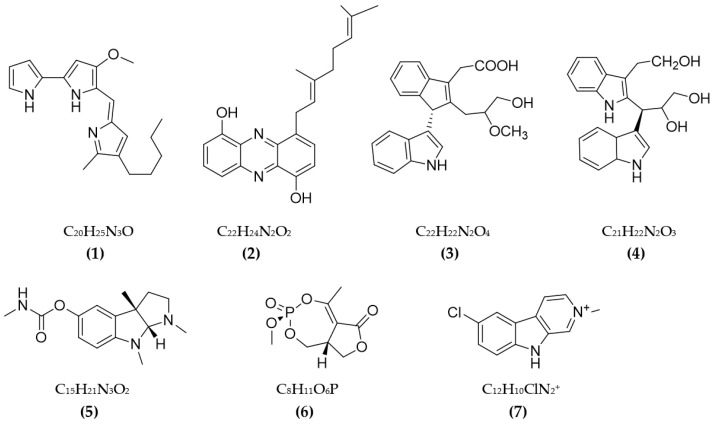
The chemical structures and chemical formula of ChE inhibitors from bacteria.

**Figure 3 pharmaceuticals-16-00580-f003:**
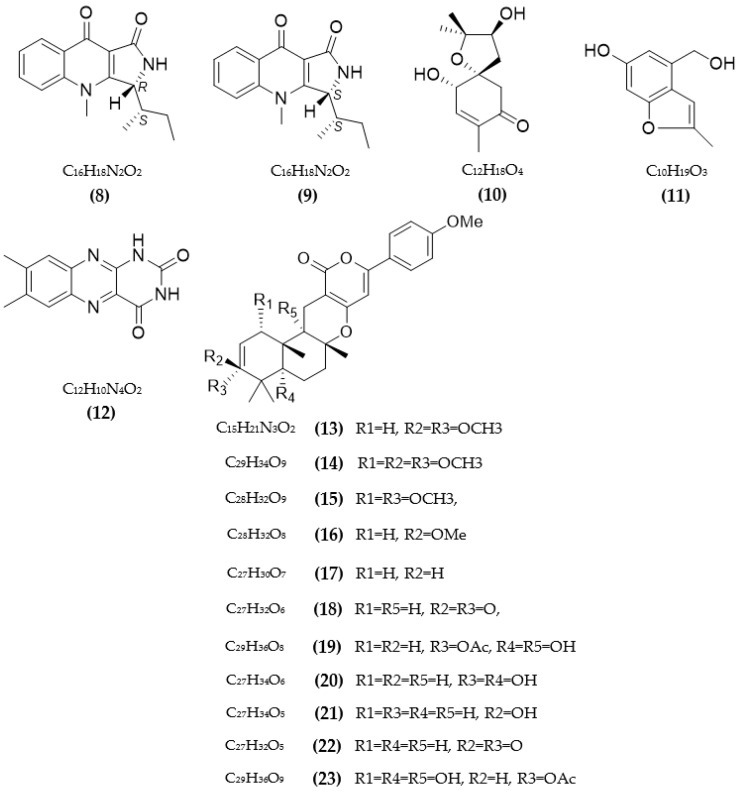
The chemical structures and chemical formula of ChE inhibitors from fungi.

**Figure 4 pharmaceuticals-16-00580-f004:**
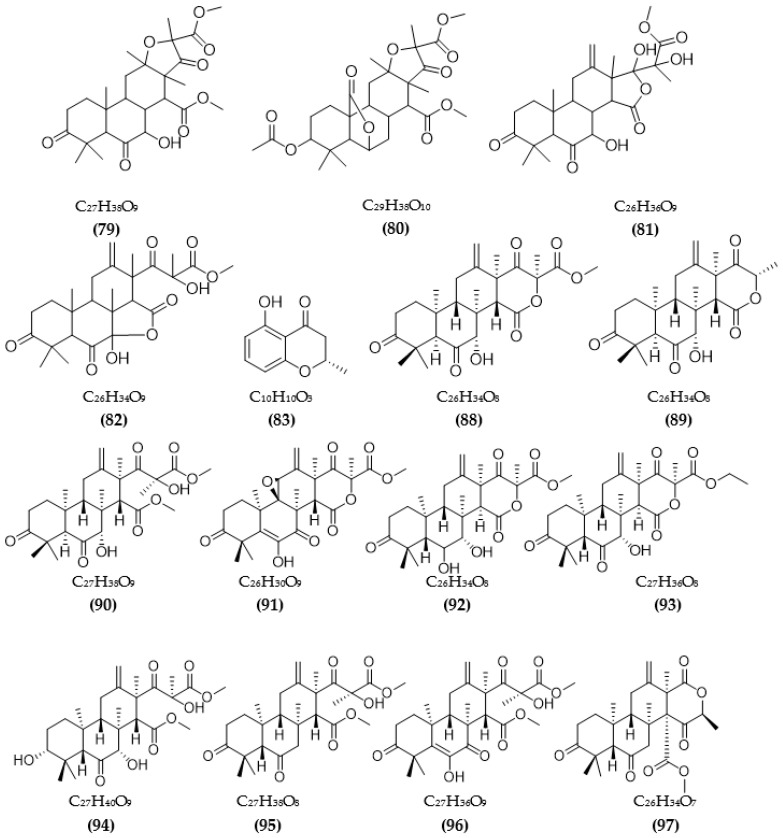
The chemical structures and chemical formula of inhibitors for other enzyme targets in AD from microorganisms.

**Table 1 pharmaceuticals-16-00580-t001:** The cholinesterase (ChE) inhibitors from bacteria.

Strain	Compound	Bioactivity (IC50)		Ref.
		AchE	BuChE	
Bacteria				
*Serratia marcescens* CC17	Prodigiosin (**1**)	640 µg/mL	ND	[62]
Actinobacteria				
*Streptomyces* sp. LB173	Geranyl-phenazine-diol (**2**)	2.62 µM	ND	[63]
*Rubrobacter radiotolerans*	2-(2-(3-hydroxy-1-(1H-indol-3-yl)-2methoxypropyl) -1H-indol-3-yl) acetic acid (**3**)	11.8 µM	ND	[64]
	3-(3-(2-hydroxyethyl)-1H-indol-2-yl) (**4**)	13.5 µM	ND	
*Streptomyces* sp. AH-14	Physostigmine (**5**)	41 μM	ND	[65]
*Streptomyces lavendulae* NK901093	Cyclophostin (**6**)	7.6 × $10^{-10}$ M	ND	[66]
*Streptomyces* sp. UTMC 1334	Pyrrole derivatives (extract)	360 µg/mL	ND	[67]
Cyanobacteria				
*Nostoc* 78-12A	Nostocarboline (**7**)	ND	13.2 μM	[68]

ND—Not determined.

**Table 2 pharmaceuticals-16-00580-t002:** The cholinesterase (ChE) inhibitors from fungi.

Strain	Compound	Bioactivity (IC50)		Ref.
		AchE	BuChE	
*Penicillium citrinum*	Quinolactacin A1 (**8**)	280 µM	-	[69]
	Quinolactacin A2 (**9**)	19.8 µM	650 µM	
*Acremonium persicinum* KUFA 1007	Acremine S (**10**)	-	-	[70]
	Acremine T (**11**)	-	-	
	Lumichrome (**12**)	12.24 µM	-	
*Penicillium* sp. FO-4259	Arisugacin (**13**)	1 nM	>18,000 nM	[71]
	Territrem B (**14**)	7.6 nM	>20,000 nM	
	Territrem C (**15**)	6.8 nM	>26,000 nM	
*Penicillium* sp. FO-4259	Arisugacin A (**16**)	0.001 nM	>21 nM	[72]
	Arisugacin B (**17**)	0.02 nM	>516 nM	
	Arisugacin C (**18**)	2.5 nM	>30 nM	
	Arisugacin D (**19**)	3.5 nM	>30 nM	
	Arisugacin E (**20**)	>100 nM	>30 nM	
	Arisugacin F (**21**)	>100 nM	>30 nM	
	Arisugacin G (**22**)	>100 nM	>30 nM	
	Arisugacin H (**23**)	>100 nM	>30 nM	
*Xylaria* sp.	Xyloketal A (**24**)	1.5 × 10^−6^ mol/L	ND	[73]
*Chrysosporium* sp.	14-(2′,3′,5′- Trihydroxyphenyl) tetradecan-2-ol (**25**)	197 µM	ND	[74]
		195 µM	ND	
		231 µM	ND	
*Paecilomyces lilacinus*	Paecilomide (**26**)	IR = 57.5%	ND	[75]
*Talaromyces* sp. LF458	Talaromycesone A (**27**)	7.49 μM	ND	[76]
	Talaroxanthenone (**28**)	1.61 μM	ND	
	AS-186c (**29**)	2.60 μM	ND	
*Syncephalastrum racemosum*	Trachyloban-19-oic acid (**30**)	>1 µM	ND	[77]
*Aspergillus sydowii*	Cyclo-(l-leu-l-pro) (**31**)	1.24 μmol/mL	ND	[78]
	Cyclo-(l-val-l-pro) (**32**)	1.06 μmol/mL	ND	
	Cyclo-(l-phe-l-val) (**33**)	1.13 μmol/mL	ND	
*Cochliobolus lunatus* SCSIO4140	Sinulariapeptide A (**34**)	1.8 µM	ND	[79]
	Sinulariapeptide B (**35**)	1.3 µM	ND	
	Phthalide glycerol (**36**)	2.5 µM	ND	
*Talaromyces aurantiacus* FL15	Asterric acid (**37**)	66.7 µM	>100 µM	[80]
	Methyl asterrate (**38**)	23.3 µM	>100 µM	
	Ethyl asterrate (**39**)	20.1 µM	>100 µM	
	Emodin (**40**)	>100 µM	>100 µM	
	Physcion (**41**)	>100 µM	>100 µM	
	Chrysophanol (**42**)	>100 µM	>100 µM	
	Sulochrin (**43**)	>100 µM	>100 µM	
*Aspergillus versicolor* Y10	Avertoxin B (**44**)	14.9 μM	ND	[81]
*Hyalodendriella* sp. Ponipodef12	Palmariol B (**45**)	115.31 µg/mL	ND	[82]
	4-hydroxymellein (**46**)	116.05 µg/mL	ND	
	Alternariol 9-methyl ether (**47**)	135.52 µg/mL	ND	
	Botrallin (**48**)	103.7 µg/mL	ND	
*Penicillium* sp. sk5GW1L	Arigsugacin I (**49**)	0.64 µM	ND	[83]
	Arigsugacin F (**50**)	0.37 µM	ND	
	Territrem B (**14**)	7.03 µM	ND	
*Penicillium chermesinum* (ZH4-E2)	3″-deoxy-6′ -O-desmethylcandidusin B (**51**)	7.8 μM	ND	[84]
	6′-O-desmethylcandidusin B (**52**)	5.2 μM	ND	
*Colletotrichum gloeosporioides*	Colletotrichine B (**53**)	38 µg/mL	ND	[85]
*Phomopsis stipata*	Koninginin T (**54**)	10.0 μg	ND	[86]
*Aspergillus flavus* LF40	Huperzine A (**55**)	0.6 μM	ND	[87]
*Amphichorda felina*	Amphichoterpenoid D (**56**)	12.5 µM	ND	[88]
	Amphichoterpenoid E (**57**)	11.6 µM	ND	
*Saccharicola* sp.	Speciosin U (**58**)	0.026 mg/mL	ND	[89]
	Trans-3,4-dihydro-3,4-dihydroxy-anofinic acid (**59**)	0.053 mg/mL	ND	
*Aspergillus terreus*	Anhydrojavanicin (**60**)	2.01 µM	ND	[90]
	8-O-methylbostrycoidin (**61**)	6.71 µM	ND	
	NGA0187 (**62**)	1.89 µM	ND	
	Beauvericin (**63**)	3.09 µM	ND	
*Aspergillus terreus*	Territrem B (**14**)	7.6 μM	ND	[91]
*Aspergillus terreus* SGP-1	Asperteretal J (**64**)	IR = 11.2%	IR = 15.6%	[92]
	Asperteretal K (**65**)	IR = 7.7%	IR = 3.2%	
	Flavipesolide B (**66**)	IR = 19.6%	IR = 22.5%	
	Butyrolactone VIII (**67**)	IR = 11.9%	IR = 67.3% K_i_ = 23.6 µM	
	Versicolactone B (**68**)	IR = 10.4%	IR = 64.4% K_i_ = 38.2 µM	
	Butyrolactone I (**69**)	-	IR = 68% K_i_ = 19.3 µM	
	Butyrolactone VII (**70**)	IR = 11.7%	IR = 76% K_i_ = 12.3 µM	
	3-hydroxy-5-[[4-hydroxy-3-(3-methyl-2-buten-1-yl) phenyl] methyl] -4- (4-hydroxyphenyl)-2(5H)-furanone (**71**)	IR = 14.2%	IR = 60.9% K_i_ = 35.7 µM	
	Butyrolactone II (**72**)	IR = 9.6%	IR = 19.9%	
	5-[(3,4-dihydro-2,2-dimethyl-2H-1-benzopyran-6-yl)methyl]-3-hydroxy-4-(4-hydroxyphenyl)-2(5H)-furanone (**73**)	IR = 23.2%	IR = 33.3%	
	Aspernolide A (**74**)	-	IR = 25.4%	
	Aspernolide B (**75**)	IR = 2.2%	IR = 13.4%	
	Aspernolide C (**76**)	IR = 3.9%	IR = 22.5%	
	Butyrolactone III (**77**)	-	-	
	Butytolactone IV (**78**)	IR = 13.3%	IR = 33.5%	
*Aspergillus terreus*	Terreusterpene A (**79**)	>40 µM	ND	[93]
	Terreusterpene B (**80**)	>40 µM	ND	
	Terreusterpene C (**81**)	>40 µM	ND	
	Terreusterpene D (**82**)	8.86 µM	ND	
*Daldinia fissa*	5-hydroxy-2-methyl-chroman-4-one (**83**)	>40 µM	>40 µM	[94]
*Aspergillus Terreus*	Territrem D (**84**)	4.2 nM	ND	[95]
	Territrem E (**85**)	4.5 nM	ND	
	Territrem B (**14**)	4.2 nM	ND	
	Territrem C (**86**)	20.1 nM	ND	
	Arisugacin A (**16**)	11.9 nM	ND	
	Arisugacin H (**23**)	5700 nM	ND	
	Terreulactone C (**87**)	50 nM	ND	

ND—Not determined (-) Inactive IR—Inhibition rate.

**Table 4 pharmaceuticals-16-00580-t004:** The docking studies of some inhibitors from microorganisms in the treatment of Alzheimer’s disease.

Strain	Enz.	Inhibitor	Experimental Activity	DS (kcal/mol)	RMSD (Å)	Linkage (Bonds)	Amino Acid Interaction	Ref.
*S.marcescens* CC17	AChE from *Electrophorus electricus* (Structure: DOI:10.2210/pdb1GQR/pdb)	Cation-prodigiosin	640 µg/mL	−12.3	1.35	6	Asp326, Asp326, Asp393, Asp393, Lys325, Asp393	[62]
		Neutral-prodigiosin		−11.1	1.75	3	Trp84, Trp84, Gly118	
*A. felina*	AChE from *Electrophorus electricus* (PDB ID: 1QTI)	Amphichoterpenoid D	12.5 µM	−9.3	ND	3	Arg289, Phe288	[88]
		Amphichoterpenoid E	11.6 µM	−9.3	ND	3	Arg28, Tyr121	
		(+)Amphichoterpenoids A	ND	−7.9	ND	2	Leu305, Glu306	
		(−)Amphichoterpenoids A	ND	−6.8	ND	0	None	
*A.sterreus* SGP-1	BChE from equine serum (EC 3.1.1.8)	Butyrolactone I	35.5 µM	−41.28	ND	>7	Gly116, Trp82, Trp231, Leu286, Val288, Phe398, Pro285, His438,	[92]
		Butyrolactone VII	18.4 µM	−48.85	ND	>7	Gly116, Trp82, Trp231, Leu286, Val288, Phe398, His438, Ala328,	
*D. fissa*	MAO-A from human recombinant (PDB ID:2Z5X)	5-hydroxy-2-methyl-chroman-4-one	13.9 μM	−6.1	ND	ND	-	[94]
	MAO-B from human recombinant (PDB ID: 4A79)		3.2 μM	−7.3	ND	ND	Cys172	
*R. corticium*	MAO-A from human recombinant (PDB ID: 2Z5X)	(S)-5-methylmellein	5.31 μM	−6.8	ND	ND	Phe208	[110]
	MAO-B from human recombinant (PDB ID: 3PO7)		9.15 μM	−6.4	ND	ND	-	
	MAO-A from human recombinant (PDB ID: 2Z5X)	(R)-5-methylmellein	ND	−6.6	ND	ND	Asn181	
	MAO-B from human recombinant (PDB ID: 3PO7)		ND	−5.2	ND	ND	-	
*Phoma* sp.	PDE4B from human	Betulinan A	44 μM	−8.071	ND	ND	Phe446	[114]
		BTH-II0204-207:A	31 μM	−8.277	ND	ND	Phe446	
		Betulinan C	17 μM	−8.732	ND	ND	Gln443, Phe446	
*B.veleznesis* RB.EK7	AChE *Electrophorus electricus* (Structure: DOI:10.2210/pdb1GQR/pdb)	Thymine	ND	−7.0	1.35	4	Asp182, Lys51, Asn183, Trp179	[132]
		Hexahydropyrrolo [1,2-a]pyra-zine-1,4-dione	ND	−6.89	1.02	3	Met175, Phe35, Lys51	

ND—No determine, IR—Inhibition rate.

## Data Availability

Not Applicable.

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
