# Peer review of "Microorganism-Derived Molecules as Enzyme Inhibitors to Target Alzheimer’s Diseases Pathways"

_pharmaceuticals, 2023, doi:10.3390/ph16040580_

Round 1

Reviewer 1 Report

This review is clear, comprehensive, and of general significance to the readers of this journal.

Focusing on Microbial Enzyme Inhibitors this review discusses the existing bibliography aiming Alzheimer's Disease (AD).

The authors begin by making a clear and synthetic mention of previous reviews on the subject, explaining the relevance of the present review in filling the gaps in the literature, namely enzyme inhibitors from microbes. The overview of AD´s pathogenesis mechanisms associated with intervenient enzymes is properly detailed and the discussion of the main enzymes is done integrating the main aspects of their bioactivity. The need to find new AD inhibitors that are safe and preferably isolated from natural sources (e.g. microbial), is also emphasized and further discussed.

The introduction part covers both old and new references in a succinct way and has perfect integration of the main aspects of the theme.

This article is well written, with a good organization of the contents and a clear and pertinent methodology, particularly the search design and strategy.

The figures and tables are appropriate to the discussion of the theme and are easy to interpret. We have only one comment to do, that´s concerning the consistency of the physical units (ml vs mL) all over the text and tables. Please fix this aspect.

The cited core references are recent and appropriate to the discussion.

The conclusions are drawn coherently and are supported by the listed citations. The research directions needed to be explored in the future are also referred, especially the exploration of inhibitors isolated from fungi.

Author Response

Reviewer # 1

This review is clear, comprehensive, and of general significance to the readers of this journal.

Focusing on Microbial Enzyme Inhibitors this review discusses the existing bibliography aiming Alzheimer's Disease (AD).

The authors begin by making a clear and synthetic mention of previous reviews on the subject, explaining the relevance of the present review in filling the gaps in the literature, namely enzyme inhibitors from microbes. The overview of AD´s pathogenesis mechanisms associated with intervenient enzymes is properly detailed and the discussion of the main enzymes is done integrating the main aspects of their bioactivity. The need to find new AD inhibitors that are safe and preferably isolated from natural sources (e.g. microbial), is also emphasized and further discussed.

The introduction part covers both old and new references in a succinct way and has perfect integration of the main aspects of the theme.

This article is well written, with a good organization of the contents and a clear and pertinent methodology, particularly the search design and strategy.

The figures and tables are appropriate to the discussion of the theme and are easy to interpret. We have only one comment to do, that´s concerning the consistency of the physical units (ml vs mL) all over the text and tables. Please fix this aspect.

The cited core references are recent and appropriate to the discussion.

The conclusions are drawn coherently and are supported by the listed citations. The research directions needed to be explored in the future are also referred, especially the exploration of inhibitors isolated from fungi.

Reply: Deeply thanks to the positive comments and your interesting view intended for our overview work. We also rechecked and fixed all of these mistakes in the revised version.

Kind regards

Reviewer 2 Report

Thi et al reviewed “Microbial Enzyme Inhibitors Targeting Alzheimer's Drugs”. This draft contains collection of inhibitors derived from microbial sources. It is an interesting collection targeting Alzheimer's disease pathways. However, few corrections need to be done.

1) Microbial Enzyme Inhibitors Targeting Alzheimer's Drugs- the current title is unclear.

suggested title is below:

Microorganism derived molecules as enzyme Inhibitors to target Alzheimer's diseases pathways.

(Or authors need to reframe the title accordingly)   

2) Figure 1- It is vague, too simple given the discovery of mechanism found over several decades. Redraw with at most details. Current figure 1 is inappropriate. The figure should depict tau pathology, aβ pathology and other associated mechanistic AD pathways. Mark the proteins that can be targeted in these pathways.

3) Are there studies for microbe derived compounds that assayed for CDK 5, ERK1/2 inhibition?

4) Add Ki column if available, in addition to IC50 in tables?

Minor

virtual study – change to in silico study

better not to mention ‘first’ in line 70 “hypothesis is the first hypothesis for AD”. Rephrase the sentence

the sentence in line 60 is vague “the determination of enzyme targets in related experimental research”. What does “related” mean here?

Use the same abbreviation throughout MAO-A or MAOA

In line 466, specify the “beta” secretase.

Author Response

Reviewer # 2

Thi et al reviewed “Microbial Enzyme Inhibitors Targeting Alzheimer's Drugs”. This draft contains collection of inhibitors derived from microbial sources. It is an interesting collection targeting Alzheimer's disease pathways. However, few corrections need to be done.

 1) Microbial Enzyme Inhibitors Targeting Alzheimer's Drugs- the current title is unclear. suggested title is below: Microorganism derived molecules as enzyme Inhibitors to target Alzheimer's diseases pathways. (Or authors need to reframe the title accordingly)   

Reply: We’ve checked and decided to revise the title of paper as your suggestion. Thanks for your advice.

2) Figure 1- It is vague, too simple given the discovery of mechanism found over several decades. Redraw with at most details. Current figure 1 is inappropriate. The figure should depict tau pathology, aβ pathology and other associated mechanistic AD pathways. Mark the proteins that can be targeted in these pathways.

Reply: Thanks for your comment. We revised Figure 1 for more clear as your suggestion.

3) Are there studies for microbe derived compounds that assayed for CDK 5, ERK1/2 inhibition?

Reply: The latest review about CDK inhibitors “Łukasik, P.; Baranowska-Bosiacka, I.; Kulczycka, K.; Gutowska, I. Inhibitors of Cyclin-Dependent Kinases: Types and Their Mechanism of Action. Int. J. Mol. Sci. 2021, 22, 2806. https://doi.org/10.3390/ijms22062806” overviewed CDK inhibitors from chemical synthesis or natural sources such as plant or marine organisms but no report mention microbial sources. The another review paper “Li T, Wang N, Zhang T, Zhang B, Sajeevan TP, Joseph V, Armstrong L, He S, Yan X, Naman CB. A Systematic Review of Recently Reported Marine Derived Natural Product Kinase Inhibitors. Mar Drugs. 2019 Aug 23;17(9):493.” about kinase inhibitor from different natural source also no mention about these two kinds of kinase enzyme. We’ve searched carefully but can not find out CDK5 and ERK1/2 inhibitors from the microorganisms. The information about ERK1/2 as an enzyme target in AD treatment was added in the revised version. Thanks for your comment.

4) Add Ki column if available, in addition to IC50 in tables?

Reply: We added Ki values in the revised version. Thanks for your suggestion.

Minor

virtual study – change to in silico study

Reply: We revised it as your mention.

better not to mention ‘first’ in line 70 “hypothesis is the first hypothesis for AD”. Rephrase the sentence

Reply: We revised it as your mention.

the sentence in line 60 is vague “the determination of enzyme targets in related experimental research”. What does “related” mean here?

Reply: A phrase was added to clarify its meaning.

Use the same abbreviation throughout MAO-A or MAOA

Reply: We revised into MAO- for all.

In line 466, specify the “beta” secretase.

Reply: We revised it as your mention. One more time, we would like to pay our deeply thanks for your very careful review, and valuable comments for the enhancement of the quality of the manuscript.

Reviewer 3 Report

The Authors describe the inhibitory profile of natural molecules from microorganisms for the treatment of Alzheimer's disease. In fact most of them showed to target notable enzymes involved in AD pathogenesis. A lot of data are presented which have required an accurate analysis of literature in this regard.

However the current presentation of data is less informative.

My first concern is about the data organisation  which are presented as a mere list of natural compounds and the related activity on one enzyme or more. No mention on the chemistry of these natural compounds and/or the main chemical determinats for the observed activity is proposed. Is there space for the identification of common chemical motifs (chemical substructure) among the investigated compounds with respect to the relevant targets? What can we learn from the chemistry point of you? A such information is complete missing. I'd introduce some pictures, accordingly.

What natural molecules have reached an advantage stage of the biological profile? Where appropriate, give more emphasis about this aspect.

The unit measure of compounds in Table 1 has to be uniformed. Please convert mg/mL into microM for a comparative purpose. 

In the section "Trends in using Virtual Screening.." the authors should first describe for each target the main residues of the enzyme active sites involved in drug interaction, using conventional drugs or known chemical compounds to map the enzyme pockets. When defined, they should speck about the chemical interactions of the selected compounds with their targets, as proposed in the main text. A comparative analysis could be yielded.

Moreover, they should define for what compounds the activity against a putative enzyme was confirmed by experimental tests. This should be an added value to the promising profile of the compounds and the metodology used for their selection.

I suggest an extensive reorganization of the paper before its acceptance.

Author Response

Reviewer # 3

The Authors describe the inhibitory profile of natural molecules from microorganisms for the treatment of Alzheimer's disease. In fact most of them showed to target notable enzymes involved in AD pathogenesis. A lot of data are presented which have required an accurate analysis of literature in this regard.

However the current presentation of data is less informative.

My first concern is about the data organization which are presented as a mere list of natural compounds and the related activity on one enzyme or more. No mention on the chemistry of these natural compounds and/or the main chemical determinants for the observed activity is proposed. Is there space for the identification of common chemical motifs (chemical substructure) among the investigated compounds with respect to the relevant targets? What can we learn from the chemistry point of you? A such information is complete missing. I'd introduce some pictures, accordingly. What natural molecules have reached an advantage stage of the biological profile? Where appropriate, give more emphasis about this aspect.

Reply: The chemical structures and chemical formula of all inhibitor compounds were drawn and added. The discussion concerning chemical-bioactivity was also added. Deeply thanks your comments to enhance the quality of manuscript.

The unit measure of compounds in Table 1 has to be uniformed. Please convert mg/mL into microM for a comparative purpose. 

Reply: Above item was revised accordingly the comments.

In the section "Trends in using Virtual Screening. the authors should first describe for each target the main residues of the enzyme active sites involved in drug interaction, using conventional drugs or known chemical compounds to map the enzyme pockets. When defined, they should speak about the chemical interactions of the selected compounds with their targets, as proposed in the main text. A comparative analysis could be yielded.

Reply: Thanks for your direction comment. Read your above suggestion, we carefully more checking the available reports about virtual study concerning enzyme inhibitors from microbes for addressing the comments. However, the papers concerning this aspect are quite few, in addition, almost these studies simple presenting some parameters of docking study, in addition, they did not used the same sorftware, as such, the comparision of some parameters was also difficult. Thus, we mentioned these docking study (and almost the parameters were described) just to emphasize an aspect is still few explored and thus maybe potential for further investigation in future as a trend. Your valuable comments with huge works should be further conducted (deeply in computation science) and may be developed as an independent Work with our suggested tittles “Trends in using Virtual Screening to Discover Potential Alzheimer's Inhibitors from Natural Products”, thus, may get enough data and for address this aspect.

In this overview work, we already do our best to addressed some comments as shown in red coulor, however, we sorry cannot more conduct virtual performance to address the data according to this comment. Thanks to give us a valuable suggested idea for our further investigation.

Moreover, they should define for what compounds the activity against a putative enzyme was confirmed by experimental tests. This should be an added value to the promising profile of the compounds and the methodology used for their selection. I suggest an extensive reorganization of the paper before its acceptance.

Reply: We’ve added bioactivity values by experimental tests to Table and address your above comments. One more time, we would like to pay our deeply thanks for your very careful review, and valuable comments for the enhancement of the quality of the manuscript.

Round 2

Reviewer 3 Report

The Authors have improved the quality of the paper, addressing most of the suggestions raised by the Reviewers. The chemistry of natural compounds is now reported and discussed. Sometimes typos and errors occur in the paper main text; make a careful check.